# Whole-Chromosome Karyotyping of Fetal Nucleated Red Blood Cells Using the Ion Proton Sequencing Platform

**DOI:** 10.3390/genes13122257

**Published:** 2022-11-30

**Authors:** Angela N. Barrett, Zhouwei Huang, Sarah Aung, Sherry S. Y. Ho, Nur Syazana Roslan, Aniza P. Mahyuddin, Arijit Biswas, Mahesh Choolani

**Affiliations:** 1Department of Obstetrics & Gynaecology, Yong Loo Lin School of Medicine, National University of Singapore, 1E Kent Ridge Road, NUHS Tower Block, Level 12, Singapore 119228, Singapore; 2iGene Laboratory Pte Ltd., 1 Science Park Road #04-10, The Capricorn, Singapore 117528, Singapore; 3Department of Obstetrics & Gynaecology, National University Hospital, 1E Kent Ridge Road, NUHS Tower Block, Level 12, Singapore 119228, Singapore

**Keywords:** fetal erythroblast, aneuploidy, single-cell whole-genome sequencing, semiconductor sequencing, cell based, non-invasive prenatal diagnosis

## Abstract

The current gold standard for the definitive diagnosis of fetal aneuploidy uses either chorionic villus sampling (CVS) or amniocentesis, both of which are which are invasive procedures carrying a procedure-related risk of miscarriage of up to 0.1–0.2%. Non-invasive prenatal diagnosis using fetal nucleated red blood cells (FNRBCs) isolated from maternal peripheral venous blood would remove this risk of miscarriage since these cells can be isolated from the mother’s blood. We aimed to detect whole-chromosome aneuploidies from single nucleated fetal red blood cells using whole-genome amplification followed by massively parallel sequencing performed on a semiconductor sequencing platform. Twenty-six single cells were picked from the placental villi of twelve patients thought to have a normal fetal genotype and who were undergoing elective first-trimester surgical termination of pregnancy. Following karyotyping, it was subsequently found that two of these cases were also abnormal (one trisomy 15 and one mosaic genotype). One single cell from chorionic villus samples for two patients carrying a fetus with trisomy 21 and two single cells from women carrying fetuses with T18 were also picked. Pooled libraries were sequenced on the Ion Proton and data were analysed using Ion Reporter software. We correctly classified fetal genotype in all 24 normal cells, as well as the 2 T21 cells, the 2 T18 cells, and the two T15 cells. The two cells picked from the fetus with a mosaic result by CVS were classified as unaffected, suggesting that this was a case of confined placental mosaicism. Fetal sex was correctly assigned in all cases. We demonstrated that semiconductor sequencing using commercially available software for data analysis can be achieved for the non-invasive prenatal diagnosis of whole-chromosome aneuploidy with 100% accuracy.

## 1. Introduction

Combined first-trimester screening for trisomies 21, 18, and 13 is routinely offered to all pregnant women, regardless of age, in many countries [1,2,3]. The addition of extra ultrasound markers at the time of a nuchal translucency scan increases the detection rate to 96% and decreases the false-positive rate to 2.5% [4,5]. Cell-free fetal DNA (cffDNA) found circulating in maternal plasma [6] represents the entire fetal genome [7] and can be used to screen for fetal aneuploidy [8,9]. Non-invasive prenatal screening (NIPS) for the three common trisomies is becoming increasingly routine as a screening test in many countries [10] and is extremely accurate [11,12]; there has been an increasing uptake of NIPS for additional chromosomal abnormalities, although with much lower positive predictive value [13,14]. Despite the increased accuracy, a positive NIPS result still requires diagnostic confirmation due to the possibility of confined placental mosaicism leading to a false positive [15,16]. For the definitive diagnosis of fetal aneuploidy, the current gold standard is to use genomic DNA (gDNA) from either a CVS or amniocentesis, both of which are invasive procedures carrying a small risk of procedure-related miscarriage [17,18]. Although this risk is lower than was previously thought [19], several studies have indicated that women would prefer prenatal tests with no associated chance of miscarriage whatsoever [20].

A major goal in prenatal diagnosis is to carry out diagnosis non-invasively using fetal cells isolated from a maternal peripheral venous blood (referred to as maternal blood from here on) sample [21]. Fetal nucleated red blood cells are ideal candidates for the cell type to be used since they are present in maternal circulation from early in the first trimester of pregnancy [22,23,24,25], have a limited lifespan (thus being pregnancy-specific), and have a unique molecular identity for isolation [21,26]. Most importantly, they are derived from the fetus and so represent the true fetal genome rather than the placental genome. Isolation of these cells remains challenging due to their scarcity in the maternal circulation, and so for test optimisation we used FNRBCs isolated from maternal villi, which are more easily accessible. Using these cells, we previously demonstrated that whole-genome amplification (WGA) followed by whole-genome sequencing (WGS) can be used to diagnose trisomies 21, 18, and 15, as well as to identify fetal sex [27].

In this study, we extended our analysis, using a semiconductor sequencing platform, to demonstrate that whole-chromosome karyotyping from single FNRBCs from both villi and post-termination of pregnancy (TOP) maternal blood can be achieved. We used the commercially available Ion Reporter software, demonstrating that it is not necessary for a laboratory to use complicated algorithms to perform the analysis.

## 2. Methods

### 2.1. Sample Collection and Processing

This study was approved by the National Healthcare Group Domain Specific Review Board, Singapore (D/07/394), and all patients provided written informed consent prior to tissue collection. Briefly, twenty-four single FNRBCs were picked manually from the placental villi of ten patients with normal fetal karyotype undergoing elective first-trimester surgical TOP at 8 + 0 to 10 + 3 weeks’ gestation (N1–N22, N25, and N26) as described previously [27]. Additionally, two single cells each were also picked from placental villi from one woman carrying a fetus with trisomy 15 (T15_1 and T15_2) and one carrying a fetus with a mosaic karyotype (47,XY,t(15;18)(p10;q10),+18(4)/46,XY(16)) (N23 and N24). All karyotypes were confirmed by QF-PCR and G-banded karyotyping [27]. One single cell from CVS samples from two patients carrying fetuses with trisomy 21 (T21_1 and T21_2) and two single cells from one woman carrying a fetus with T18 (T18_1 and T18_2) were also picked, yielding a total of six trisomic cells (gestational ages between 8 + 1 and 13 + 2 weeks).

Six cells were isolated from the maternal blood of two patients post-TOP, as described previously [27].

### 2.2. Massively Parallel Sequencing

All single cells were subjected to whole-genome amplification (WGA) using PicoPLEX (Rubicon Genomics, Ann Arbor, MI, USA). The procedure was carried out according to the manufacturer’s instructions, with an additional extension of 72 °C for 20 min at the end. WGA products were purified using an ISOLATE II PCR and Gel kit (Bioline, Meridian Bioscience Asia Pte Ltd., Singapore). Following WGA, libraries for WGS were prepared using the Ion Plus Fragment Library Kit (Thermo Fisher Scientific) according to the manufacturer’s instructions, with addition of indexes for each sample. Samples were diluted to 100 pM and pooled in batches of 16. Pooled DNA was loaded onto the Ion One Touch 2 System (Thermo Fisher Scientific) at a concentration of 45 pM, and library amplification and enrichment were carried out according to the manufacturer’s instructions. Sequencing was performed on the Ion Proton (Thermo Fisher Scientific) using a v3 Ion PI chip.

### 2.3. Data Analysis

FASTQ data generated by the Ion Proton were assessed using FASTQC (https://www.bioinformatics.babraham.ac.uk/projects/fastqc/ (accessed on 20 January 2019)) to check the quality. A minimum threshold of 28 was used to indicate good-quality bases. Upon passing the QC check, FASTQ files were aligned to the human hg19 reference genome, and then duplicate reads were removed using Samtools. The deduplicated BAM files were uploaded to the Ion Reporter 5.0 software (Thermo Fisher Scientific). Aneuploidy calls were automatically generated using the low-pass whole-genome aneuploidy workflow (https://ionreporter.thermofisher.com/ionreporter/help/GUID-4A207E04-2113-4633-968D-4B7A65A1D64A.html (accessed on 20 January 2019)). The software generates a median of the absolute values of all pairwise differences (MAPD) score, which is a measure of the noise seen for a region or genome. One of the criteria for making a CNV call is that the MAPD score must be <0.4.

## 3. Results

Data from all samples passed the FASTQC check. We achieved a median of 3,749,825 unique reads per single-cell sample (interquartile range: 3,319,223–4,425,936). All 24 normal post-TOP placental villi cells were confirmed to be unaffected using non-invasive prenatal diagnosis (NIPD) (Table 1). The fetal sex was also correctly assigned to each sample. The signal for each chromosome was clear, with very little noise in the whole-genome plots. Representative plots are shown in Figure 1a (a normal male) and b (a normal female). The two T21 cells, the two T18 cells, and the two T15 cells all showed the expected chromosome gain, with all other chromosomes showing a diploid signal (Figure 1c–e).

We tested two cells from a fetus predicted by CVS to have a mosaic karyotype and instead found a normal male genotype for both cells, indicating that this could probably be a case of confined placental mosaicism (Figure 1f). One of the six single cells isolated from post-TOP maternal blood (sample MB150-1c3) failed quality control due to insufficient reads and so was excluded from further analysis. We obtained a normal genotype for all five remaining cells from the two cases of maternal blood post-TOP, as expected (Figure 1g,h). Despite this slight additional noise, the sample quality for the single FNRBCs fell within the acceptable range, and the results were very clearly normal. Using Ion Reporter to call a CNV, the MAPD score should be below 0.4; for the two FNRBCs from maternal blood shown in Figure 1g,h, the MAPD values were 0.222 and 0.197, respectively.

## 4. Discussion

We demonstrated that whole-chromosome karyotyping from single FNRBCs from both villi and post-TOP maternal blood can be achieved using a semiconductor sequencing platform and analysed using commercially available Ion Reporter software. Thus, we successfully showed that it is not necessary for a laboratory to use complicated algorithms to perform the analysis.

Placental tissue is often used as a surrogate to screen or diagnose fetal well-being. Both CVS and NIPS involve obtaining fetal DNA from the placenta through invasive and non-invasive procedures, respectively. This has given rise to the problematic cases of false-positive and false-negative results due to confined placental mosaicism. A retrospective audit of >50,000 CVS samples sent for cytogenetic analyses determined that the frequency of false positives due to CPM ranged from 1/1065 to 1/3931 cases, whereas false negatives occurred in 1/107 [28]. Moreover, NIPS false-positive and false-negative test results have been highlighted by the US FDA in a recent safety communication [29]. There have been several recent studies demonstrating successful cell-based NIPD using trophoblasts from maternal blood [30,31,32,33] or from the cervix [34,35,36], which are present in the circulation at higher numbers, are easier to isolate, and are free of maternal DNA contamination; however, these studies do not totally avoid CPM [30,31]. Our results serve to highlight the advantage of FNRBCs over trophoblasts, namely, that FNRBCs represent the true fetal genotype, originating from the yolk sac and then from fetal liver at later gestations [37], and can be found within first-trimester placental villi, where they undergo terminal maturation processes [38], whereas trophoblasts are derived from the placenta. In the case of confined placental mosaicism, a proportion of cells will be aneuploid and the rest normal. Since only a few fetal cells will be available for analysis, trophoblasts have the potential to yield a false-positive result if only the aneuploid cells from the mosaic placenta happen to be picked. This result would be extremely problematic since this method is intended as a definitive diagnosis. In our mosaic trisomy 18 sample, the genotype revealed normal male from both single FNRBCs, whilst the CVS karyotype reported two cell lines: mosaic 47,XY,t(15;18)(p10;q10),+18[4]/46,XY[16] and normal male. Therefore, a much larger number of FNRBCs (i.e., 20 cells) has to be tested before a definitive diagnosis of fetal mosaicism can be achieved.

The whole-chromosome genotyping signals for all cells isolated from maternal blood were noisier than those for the placental villi cells. Two factors may contribute to this. Firstly, a longer procedure is required for the isolation of cells from maternal blood compared with the isolation from villi, possibly affecting the quality of the obtained gDNA. Secondly, the FNRBCs in the maternal circulation will range in age from those freshly released during the TOP procedure and those which have been circulating since earlier in the pregnancy; the older cells may have poorer-quality DNA [39]. Therefore, we can still have confidence in the data generated from these cells. It is possible that further improvements to the genotype signal may be achieved using a cell-type-specific reference set (i.e., a reference made up from single FNRBCs isolated from post-TOP maternal blood from females carrying fetuses with normal genotypes) rather than the default gDNA reference used in the Ion Reporter software.

The most important question is whether the same reliable results can be accomplished in FNRBCs from maternal blood in ongoing pregnancies. It was demonstrated that at least one FNRBC per ml of maternal blood can be isolated [40], and work is ongoing in our lab and others to develop robust methods for capturing these rare cells reliably [24,41,42,43]. In the future, we hope to extend our analysis further to include microdeletions and microduplications. These abnormalities occur in 1%–1.7% of all pregnancies, a much higher frequency than trisomy 21 [44]. Although some microdeletions and microduplications can be assessed using NIPS, the false-positive rate is high [13,45], and so using fetal cells for non-invasive prenatal diagnosis in families at risk or following identification of a structural abnormality by ultrasound would help to reduce the need for invasive testing. Deeper sequencing or targeted methods will be necessary to identify these copy number changes, as has been demonstrated using cell-free DNA [46].

## 5. Conclusions

We demonstrated that we can use semiconductor sequencing for non-invasive prenatal diagnosis of whole-chromosome aneuploidy in FNRBCs, with 100% concordance in 31 fetal cells passing QC metrics. Although signals from post-TOP maternal blood were a little chaotic, the normal genotypes were still called very clearly by the software, falling within acceptable quality ranges. Once we have optimised protocols for the isolation of the target cells, we hope that this method will be readily translated into clinical practice.

## Figures and Tables

**Figure 1 genes-13-02257-f001:**
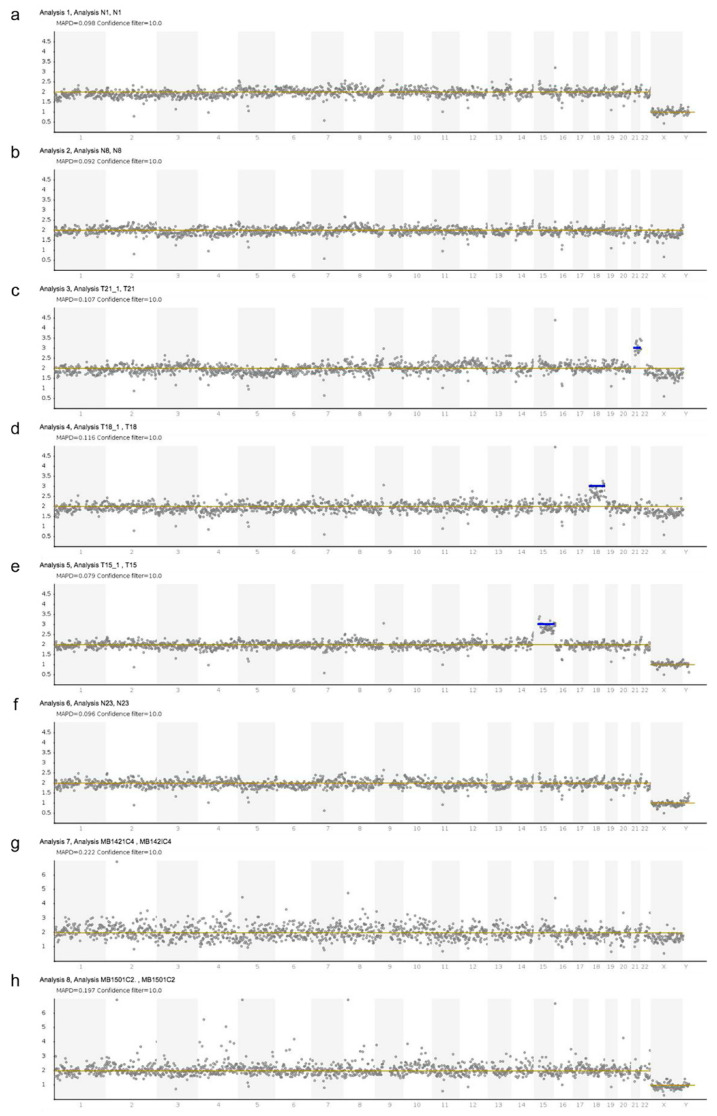
Representative whole-genome plots for different whole-chromosome copy number analyses or genotyping from single FNRBCs. Normal genotypes obtained for single cells from (**a**) a normal male fetus and (**b**) a normal female. Aneuploid cells were tested from (**c**) a T21 female, (**d**) a T18 female, and (**e**) a T15 male; all showed the expected chromosomal gains with low signal noise for the normal chromosomes. (**f**) A single cell from a normal male fetus in a patient with confined placental mosaicism (CPM). Two cells from post-TOP maternal blood: (**g**) a normal female fetus and (**h**) a normal male. Aneuploidies are indicated by a blue bar. MAPD values are indicated for each sample, acting as a sample quality metric. Values above 0.3 are unacceptable.

**Table 1 genes-13-02257-t001:** Details of the analysed samples. MB—maternal blood, TOP—termination of pregnancy, NIPD—non-invasive prenatal diagnosis, MAPD—median absolute pairwise difference, CVS—chorionic villus sampling.

Sample	Sample Type	Karyotype	Gestation (Weeks)	NIPD Genotype	NIPD Fetal Sex	MAPD
N1	Post-TOP villi	46,XY	8 + 2	Normal	M	0.098
N2	Post-TOP villi	46,XY	8 + 2	Normal	M	0.092
N3	Post-TOP villi	46,XX	8 + 1	Normal	F	0.103
N4	Post-TOP villi	46,XX	8 + 1	Normal	F	0.088
N5	Post-TOP villi	46,XX	8 + 2	Normal	F	0.095
N6	Post-TOP villi	46,XX	8 + 2	Normal	F	0.096
N7	Post-TOP villi	46,XX	9 + 0	Normal	F	0.094
N8	Post-TOP villi	46,XX	9 + 0	Normal	F	0.092
N9	Post-TOP villi	46,XY	8 + 1	Normal	M	0.087
N10	Post-TOP villi	46,XY	8 + 1	Normal	M	0.098
N11	Post-TOP villi	46,XX	8 + 3	Normal	F	0.096
N12	Post-TOP villi	46,XX	8 + 3	Normal	F	0.106
N13	Post-TOP villi	46,XY	10 + 0	Normal	M	0.091
N14	Post-TOP villi	46,XY	10 + 0	Normal	M	0.102
N15	Post-TOP villi	46,XX	8 + 0	Normal	F	0.102
N16	Post-TOP villi	46,XX	8 + 0	Normal	F	0.110
N17	Post-TOP villi	46,XX	8 + 4	Normal	F	0.098
N18	Post-TOP villi	46,XX	8 + 4	Normal	F	0.100
N19	Post-TOP villi	46,XX	8 + 2	Normal	F	0.106
N20	Post-TOP villi	46,XX	9 + 0	Normal	F	0.100
N21	Post-TOP villi	46,XX	8 + 4	Normal	F	0.107
N22	Post-TOP villi	46,XX	8 + 4	Normal	F	0.100
N23	Post-TOP villi	47,XY,t(15;18) (p10;q10),+18(4)/46,XY(16)	8 + 6	Normal	M	0.096
N24	Post-TOP villi	47,XY,t(15;18) (p10;q10),+18(4)/46,XY(16)	8 + 6	Normal	M	0.098
N25	Post-TOP villi	46,XY	8 + 1	Normal	M	0.105
N26	Post-TOP villi	46,XY	8 + 4	Normal	M	0.099
T21_1	CVS villi	47,XX,+21	12 + 0	T21	F	0.107
T21_2	CVS villi	47,XY,+21	13 + 0	T21	M	0.099
T18_1	CVS villi	47,XX,+18	12 + 4	T18	F	0.116
T18_2	CVS villi	47,XX,+18	12 + 4	T18	F	0.133
T15_1	Post-TOP villi	47,XY,+15	8 + 1	T15	M	0.079
T15_2	Post-TOP villi	47,XY,+15	8 + 1	T15	M	0.090
MB142_1c2	Post-TOP MB	46,XX	8 + 2	Normal	F	0.224
MB142_1c3	Post-TOP MB	46,XX	8 + 2	Normal	F	0.201
MB142_1c4	Post-TOP MB	46,XX	8 + 2	Normal	F	0.222
MB150_1c1	Post-TOP MB	46,XY	9 + 0	Normal	M	0.171
MB150_1c2	Post-TOP MB	46,XY	9 + 0	Normal	M	0.197
MB150_1c3	Post-TOP MB	46,XY	9 + 0	Normal	M	1.251

## Data Availability

Not applicable.

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
