# Peer review of "Whole-Chromosome Karyotyping of Fetal Nucleated Red Blood Cells Using the Ion Proton Sequencing Platform"

_genes, 2022, doi:10.3390/genes13122257_

Round 1

Reviewer 1 Report

This is a study investigating aneuploidy for chromosomes 21, 18 and 15 in a modest sampling of pregnancies using nucleated fetal red blood cells recovered from chorionic villi as a source, with genome-wide amplification of the DNA in individual fetal cells and massively parallel sequencing using the Ion Proton platform.

The paper reads reasonably well and mostly clear language.  The sections are not properly organized, however. This work will likely be of interest to readers of this journal, and it does not appear to duplicate other work already known in the field.  The work appears to have been properly vetted for ethical considerations. I did not identify any serious scientific flaws that made me question the soundness of the work.

My concerns are as follows:

1) the literature that is cited is not always appropriate.  For example, the 2nd citation occurs in the context of making a point about current practice in prenatal screening, but the citation is badly outdated.  There are any number of more recent articles to make this point, and I would suggest something less than three years old as this area of study is changing fairly rapidly.  References #7 and #8 cite the use of cfDNA in prenatal screening practice, but the date on the articles cited (2008) predate clinical trials for this technology and approach.  Later in the paper, the authors cite a much larger number of studies than is needed to make their points (e.g., references 37-43 are reinforcing a single simple point; similarly for references 27-36; and references 46-54). The authors should select the smallest number of references that adequately document the foundational science required to make their argument.

2) Much of what appears in the Results Section belongs in the Discussion Section, and the Discussion section is absurdly short as a result. As a rule of thumb, if the authors feel compelled to cite literature, in all likelihood, they have moved from presenting actual results, to discussing the significance of the findings.  The organization of these two sections requires some reorganization.

3) In various places, the authors are using insufficiently precise language. This begins with the misrepresentation of "karyotyping" (which is a cytogenetic procedure of sorting of the chromosomes according to number) when in fact they are performing whole chromosome copy number analysis.  The authors do not describe that they have done any karyotyping , although they describe confirmation of case findings and a detailed karyotypic result for a mosaic trisomy 18 with a balanced reciprocal translocation (also see note below about this case). I am sympathetic to the idea that copy number analysis that can be performed by NGS is likely to obviate the need for the direct study of chromosomes under a microscope, but it is not itself karyotyping. Elsewhere in the paper, there is use of the abbreviation NIPD, which presumably is intended to mean "non-invasive prenatal diagnosis".  The literature is notoriously confused on how best to refer to the use of cell-free fetal DNA in maternal circulation for prenatal screening.  The American College of Medical Genetics & Genomics (ACMG) has discouraged use of the term "non-invasive prenatal testing" (NIPT) in favor of "non-invasive prenatal screening" (NIPS) as the former term has caused clinical confusion among patients and their providers as to whether or not the findings are diagnostic (as the authors point out, they are not).  This specific kind of confusion undermines the authors own assertion that nucleated fetal RBCs are fetal in origin (rather than extra-embryonic as are chorionic cells) and therefore would be expected to be diagnostic.  My suggestion is to adopt the term NIPS for the use of cfDNA prenatal screening and to make explicit any claims that FNRBCs are more likely to turn out to be diagnostic (which stands to reason, but certainly has not been demonstrated through any large clinical trials).  This point, that FNRBCs are fetal in origin is perhaps confused by the procedure used to obtain them (extracting them from the extra-embryonic villi), and I think their argument might be strengthened to briefly explain why cells of fetal origin are found in the villi.

4) I noted some conceptual confusion in the conclusions based on the study of the mosaic fetus. Each cell in a mosaic fetus has either the normal chromosomal complement or the abnormal.  In selecting just one or two cells (two in this case), one cannot make any claims about mosaicism as it would require a much larger sample size (typically twenty for prenatal diagnosis using CVS or amnio).  The more obvious interpretation is that the two cells selected for NFRBC study happened to be from the normal cell line.  Even though you might have 4 million+ data points from NGS, if they all originated from the same cell, the relevant sample size is n=1. 

5) It is possible that the karyotype of the mosaic embryo is in fact 47,XY,t(15;18),+18[4]/46,XY[16], but it seems highly likely that the trisomy 18 resulted from a 3:1 segregation of a parental balanced t(15;18).  If so, the second cell line (the one with 16 counts) would more likely be 46,XY,t(15;18) rather than 46,XY.  That should be verified with the cytogenetics laboratory as it may rather be 47,XY,t(15;18)(p10;q10)+18[4]/46,XY,t(15;18((p10;q10)[16].  That second cell line would appear to have a normal chromosome complement using NGS because the rearrangement is balanced.

6) Did the authors mean to include the journal instructions immediately following the Discussion Section?

Author Response

Reviewer 1

This is a study investigating aneuploidy for chromosomes 21, 18 and 15 in a modest sampling of pregnancies using nucleated fetal red blood cells recovered from chorionic villi as a source, with genome-wide amplification of the DNA in individual fetal cells and massively parallel sequencing using the Ion Proton platform.

The paper reads reasonably well and mostly clear language.  The sections are not properly organized, however. This work will likely be of interest to readers of this journal, and it does not appear to duplicate other work already known in the field.  The work appears to have been properly vetted for ethical considerations. I did not identify any serious scientific flaws that made me question the soundness of the work.

We would like to thank the reviewer for his/her careful review and thoughtful comments of our manuscript. We would also like to thank the reviewer for recognising the relevance and importance of our paper in the area of cell-based non-invasive prenatal diagnosis. We a grateful for the opportunity to address the reviewers’ concerns.

  • the literature that is cited is not always appropriate.  For example, the 2nd citation occurs in the context of making a point about current practice in prenatal screening, but the citation is badly outdated.  There are any number of more recent articles to make this point, and I would suggest something less than three years old as this area of study is changing fairly rapidly. 

We thank the reviewer for the constructive comment. Citation 2 has been replaced with updated references as below to better reflect current practices.

  • Gadsboll, K., et al., Current use of noninvasive prenatal testing in Europe, Australia and the USA: A graphical presentation. Acta Obstet Gynecol Scand, 2020. 99(6): p. 722-730.
  • van der Meij, K.R.M., et al., TRIDENT-2: National Implementation of Genome-wide Non invasive Prenatal Testing as a First-Tier Screening Test in the Netherlands. Am J Hum Genet, 2019. 105(6): p. 1091-1101.

  • References #7 and #8 cite the use of cfDNA in prenatal screening practice, but the date on the articles cited (2008) predate clinical trials for this technology and approach.  Later in the paper, the authors cite a much larger number of studies than is needed to make their points (e.g., references 37-43 are reinforcing a single simple point; similarly for references 27-36; and references 46-54). The authors should select the smallest number of references that adequately document the foundational science required to make their argument.

We thank the reviewer for the comment on the suitability of references Chiu et al., 2008 and Fan et al., 2008 in the manuscript. These references were cited for the first reports of aneuploidy screening methods using cell-free fetal DNA rather than the current NIPS practices. We hope this provides better clarity and is acceptable to the reviewer. We would also like to thank the reviewer for highlighting our tendency for overzealous referencing. We have reduced the number of references cited to only the pertinent ones.

References 37-43 was reduced to four references. Please see below.

  1. Breman, A.M., et al., Evidence for feasibility of fetal trophoblastic cell-based noninvasive prenatal testing. Prenat Diagn, 2016. 36(11): p. 1009-1019.
  2. Vossaert, L., et al., Validation Studies for Single Circulating Trophoblast Genetic Testing as a Form of Noninvasive Prenatal Diagnosis. Am J Hum Genet, 2019. 105(6): p. 1262-1273.
  3. Crovetti, B., et al., Circulating trophoblast numbers as a potential marker for pregnancy complications. Prenat Diagn, 2022.
  4. Afshar, Y., et al., Circulating trophoblast cell clusters for early detection of placenta accreta spectrum disorders. Nat Commun, 2021. 12(1): p. 4408.

References 27-36 was reduced to three references. Please see below.

  1. Pfeifer, I., et al., Cervical trophoblasts for non-invasive single-cell genotyping and prenatal diagnosis. Placenta, 2016. 37: p. 56-60.
  2. Bailey-Hytholt, C.M., et al., Enrichment of Placental Trophoblast Cells from Clinical Cervical Samples Using Differences in Surface Adhesion on an Inclined Plane. Ann Biomed Eng, 2021. 49(9): p. 2214-2227.
  3. Jain, C.V., et al., Fetal genome profiling at 5 weeks of gestation after noninvasive isolation of trophoblast cells from the endocervical canal. Sci Transl Med, 2016. 8(363): p. 363re4.

References 46-54 was reduced to three references. Please see below.

  1. Huang Z, C.C., Mahyuddin AP, Liu Y, Wong CC, Choolani M, Isolation of fetal nucleated red blood cells using a microfilter chip for non-invasive prenatal diagnosis. Prenat Diagn, 2014. 34: p. 22-86.
  2. Kadam P, P.S., Zhang H, Mahyuddin AP, Ismail NS, Shikkander N, et al. , A novel marker for isolation of fetal nucleated red blood cells for non-invasive prenatal diagnosis. Prenat Diagn, 2012. 32: p. 1-128.
  3. Wang, Z., et al., Enhanced Isolation of Fetal Nucleated Red Blood Cells by Enythrocyte-Leukocyte Hybrid Membrane-Coated Magnetic Nanoparticles for Noninvasive Pregnant Diagnostics. Anal Chem, 2021. 93(2): p. 1033-1042.
  • Much of what appears in the Results Section belongs in the Discussion Section, and the Discussion section is absurdly short as a result. As a rule of thumb, if the authors feel compelled to cite literature, in all likelihood, they have moved from presenting actual results, to discussing the significance of the findings.  The organization of these two sections requires some reorganization.

We thank the reviewer for the recommendation and have re-organised the Results and Discussion Sections appropriately. Kindly see amended manuscript with tracked changes. We hope the re-organisation of sections has provided better structure and readability for the readers.

4) In various places, the authors are using insufficiently precise language. This begins with the misrepresentation of "karyotyping" (which is a cytogenetic procedure of sorting of the chromosomes according to number) when in fact they are performing whole chromosome copy number analysis.  The authors do not describe that they have done any karyotyping , although they describe confirmation of case findings and a detailed karyotypic result for a mosaic trisomy 18 with a balanced reciprocal translocation (also see note below about this case).

We thank the reviewer for highlighting the use of imprecise language and have amended “karyotyping” when describing whole chromosome copy number analysis to “genotyping. Karyotyping or G-banding was performed for all villi samples as referenced in our previous publication Hua R, Barrett AN, Tan TZ, Huang Z, Mahyuddin AP, Ponnusamy S, Sandhu JS, Ho SS, Chan JK, Chong S, Quan S, Choolani M. Detection of aneuploidy from single fetal nucleated red blood cells using whole genome sequencing. Prenat Diagn. 2015 Jul;35(7):637-44. doi: 10.1002/pd.4491. Epub 2014 Oct 16. PMID: 25178640. We include the karyotype report for the mosaic trisomy 18 case for reference. “Karyotype: 47,XY,t(15;18)(p10;q10),+18[4]/46,XY[16]. Interpretation: Male karyotype with. Two cell lines. Of twenty metaphases analysed four metaphases show 47 chromosomes with a translocation between chromosomes 15 and 18., along with an extra copy of chromosome 18. Sixteen metaphases show a normal male karyotype. This has resulted in mosaic trisomy 18.”

5) I am sympathetic to the idea that copy number analysis that can be performed by NGS is likely to obviate the need for the direct study of chromosomes under a microscope, but it is not itself karyotyping. Elsewhere in the paper, there is use of the abbreviation NIPD, which presumably is intended to mean "non-invasive prenatal diagnosis". 

We thank the reviewer for spotting this and have included non-invasive prenatal diagnosis prior to the use of NIPD abbreviation.

6) The literature is notoriously confused on how best to refer to the use of cell-free fetal DNA in maternal circulation for prenatal screening.  The American College of Medical Genetics & Genomics (ACMG) has discouraged use of the term "non-invasive prenatal testing" (NIPT) in favor of "non-invasive prenatal screening" (NIPS) as the former term has caused clinical confusion among patients and their providers as to whether or not the findings are diagnostic (as the authors point out, they are not).  This specific kind of confusion undermines the authors own assertion that nucleated fetal RBCs are fetal in origin (rather than extra-embryonic as are chorionic cells) and therefore would be expected to be diagnostic.  My suggestion is to adopt the term NIPS for the use of cfDNA prenatal screening and to make explicit any claims that FNRBCs are more likely to turn out to be diagnostic (which stands to reason, but certainly has not been demonstrated through any large clinical trials).  This point, that FNRBCs are fetal in origin is perhaps confused by the procedure used to obtain them (extracting them from the extra-embryonic villi), and I think their argument might be strengthened to briefly explain why cells of fetal origin are found in the villi.

We thank the reviewer for the comment and whole heartedly agree with the reviewer regarding the use of “Non-invasive prenatal screening" (NIPS) instead of "Non-invasive prenatal testing" (NIPT) to describe cell-free fetal DNA prenatal screening. We have amended the manuscript as per recommendation (see tracked changes) and believe that this has strengthened our assertion that FNRBCs are fetal in origin and therefore, diagnostic. We have also included a brief explanation on the origins  of FNRBCs and why they can be found in first trimester placental villi in the manuscript.  Line 174-179:  Our results serve to highlight the advantage of FNRBCs over trophoblasts: namely, that FNRBCs represent the true fetal genotype, originating from the yolk sac then from fetal liver at later gestations (Palis et al., 2001) and can be found within first trimester placental villi where they undergo terminal maturation process (Van Handel et al., 2010).

  • I noted some conceptual confusion in the conclusions based on the study of the mosaic fetus. Each cell in a mosaic fetus has either the normal chromosomal complement or the abnormal.  In selecting just one or two cells (two in this case), one cannot make any claims about mosaicism as it would require a much larger sample size (typically twenty for prenatal diagnosis using CVS or amnio).  The more obvious interpretation is that the two cells selected for NFRBC study happened to be from the normal cell line.  Even though you might have 4 million+ data points from NGS, if they all originated from the same cell, the relevant sample size is n=1

We thank the reviewer for the astute interpretation of the mosaic trisomy 18 sample. We agree that a larger number of FNRBCs has to be tested before a definitive diagnosis of fetal mosaicism can be made. As such, we have added the sentences in the Discussion section. Line 182-185: In our mosaic trisomy 18 sample, the genotype revealed normal male from both single FNRBCs whilst the CVS karyotype reported two cell lines: mosaic 47,XY,t(15;18)(p10;q10),+18[4]/46,XY[16] and normal male. Therefore, a much larger number of FNRBCs (i.e. 20 cells) has to be tested before a definitive diagnosis of fetal mosaicism can be made.”

8)         It is possible that the karyotype of the mosaic embryo is in fact 47,XY,t(15;18),+18[4]/46,XY[16], but it seems highly likely that the trisomy 18 resulted from a 3:1 segregation of a parental balanced t(15;18).  If so, the second cell line (the one with 16 counts) would more likely be 46,XY,t(15;18) rather than 46,XY.  That should be verified with the cytogenetics laboratory as it may rather be 47,XY,t(15;18)(p10;q10)+18[4]/46,XY,t(15;18((p10;q10)[16].  That second cell line would appear to have a normal chromosome complement using NGS because the rearrangement is balanced.

We thank the reviewer for the insightful comment and agree that this may be a possibility. We will discuss this case further with the cytogenetics laboratory.

9) Did the authors mean to include the journal instructions immediately following the Discussion Section?

We thank the reviewer for the comment and apologise for the error in manuscript formatting. Journal instruction has been omitted.

Reviewer 2 Report

This manuscript presents an advance over previous work that showed that whole genome sequencing of hand-picked single fetal nucleated red cells can give an accurate measure of aneuploidy, including trisomy 21.  The disadvantage of standard whole genome sequencing is that it is expensive and thus not appropriate for mass use.  The current paper provides convincing evidence that a semi-conductor sequencing platform (Rubicon Genomics and Ion Reporter software) applying massively parallel sequencing of amplified DNA from 26 single cells derived from maternal blood, CVS, and fetus with know karyotypes correctly identified the aneuploidy in all cases. This finding may increase the availability non-invasive fetal karyotyping.

Author Response

Reviewer 2

This manuscript presents an advance over previous work that showed that whole genome sequencing of hand-picked single fetal nucleated red cells can give an accurate measure of aneuploidy, including trisomy 21.  The disadvantage of standard whole genome sequencing is that it is expensive and thus not appropriate for mass use.  The current paper provides convincing evidence that a semi-conductor sequencing platform (Rubicon Genomics and Ion Reporter software) applying massively parallel sequencing of amplified DNA from 26 single cells derived from maternal blood, CVS, and fetus with know karyotypes correctly identified the aneuploidy in all cases. This finding may increase the availability non-invasive fetal karyotyping.

We would like to thank the reviewer for his/her careful review and thoughtful comments of our manuscript. We would also like to thank the reviewer for recognising the relevance and importance of our paper in the area of cell-based non-invasive prenatal diagnosis

Round 2

Reviewer 1 Report

The authors did a nice job of addressing previous concerns.  I remain concerned that the karyotype description of the mosaic trisomy 18 case with a balanced translocation is incorrect. The cell line that is described as 46,XY almost certainly would have the balanced reciprocal translocation, but that is for the authors to settle with their cytogenetics laboratory.

I noticed on line 159 of the revised manuscript has a grammatical error (it should be "have" rather than "has").

Author Response

We thank the reviewer for spotting the grammatical error. We have made the amendment and the sentence now reads (Line: 186-187) “Moreover, NIPS false positive and negative test results have been highlighted by the US FDA in a recent safety communication”.
